# Harmonization of Biobank Education for Biobank Technicians: Identification of Learning Objectives

**DOI:** 10.3390/biotech10020007

**Published:** 2021-04-14

**Authors:** Mara Lena Hartung, Ronny Baber, Esther Herpel, Cornelia Specht, Daniel Peer Brucker, Anne Schoneberg, Theresa Winter, Sara Yasemin Nussbeck

**Affiliations:** 1German Biobank Node, Universitäts Medizin Charité, 13353 Berlin, Germany; mara-lena.hartung@charite.de (M.L.H.); cornelia.specht@charite.de (C.S.); 2Institute of Laboratory Medicine, Clinical Chemistry, and Molecular Diagnostics, University Hospital Leipzig, 04103 Leipzig, Germany; ronny.baber@medizin.uni-leipzig.de; 3Institute of Pathology, Heidelberg University Hospital, 69121 Heidelberg, Germany; esther.herpel@gmail.com; 4Interdisciplinary Biobank and Database Frankfurt (iBDF), University Hospital Frankfurt, 60590 Frankfurt, Germany; daniel.brucker@kgu.de; 5Central Biobank UMG and Department of Medical Informatics, University Medical Center Göttingen, 37075 Göttingen, Germany; anne.schoneberg@med.uni-goettingen.de; 6Integrated Research Biobank, University Medicine Greifswald, 17475 Greifswald, Germany; theresa.winter@med.uni-greifswald.de; 7Central Biobank UMG, University Medical Center, 37075 Göttingen, Germany

**Keywords:** advanced training, biobank education, biobank technician, learning objectives

## Abstract

The quality of biospecimens stored in a biobank depends tremendously on the technical personnel responsible for processing, storage, and release of biospecimens. Adequate training of these biobank employees would allow harmonization of correct sample handling and thus ensure a high and comparable quality of samples across biobank locations. However, in Germany there are no specific training opportunities for technical biobank staff. To understand the educational needs of the technical personnel a web-based survey was sent to all national biobanks via established e-mail registers. In total, 79 biobank employees completed the survey, including 43 technicians. The majority of the participating technical personnel stated that they had worked in a biobank for less than three years and had never participated in an advanced training. Three-quarters of the technicians indicated that they were not able to understand English content instantly. Based on these results and the results of a workshop with 16 biobank technicians, 41 learning objectives were formulated. These learning objectives can be used as a basis for advanced training programs for technical personnel in biobanks. Setting up courses based on the identified learning objectives for this group of biobank staff could contribute to harmonization and sustainability of biospecimen quality.

## 1. Introduction

Biobanks are an essential resource for biomedical research and key players in the development of novel therapies and diagnostic markers. The constant improvements in life science technologies open completely new perspectives in personalized treatment options and are one reason for an increasing demand of high-quality biospecimens and associated data [1]. By managing and distributing high-quality biospecimens and data, biobanks meet the needs of modern research, thus providing an important contribution to reliable and reproducible scientific results [2,3]. By now, an alarming number of irreproducible preclinical studies are a consequence of improper biospecimens or reference material, wasting approximately US$ 28 billion per year in the United Stated of America [4,5].

To avoid unusable biospecimens and to achieve sustainable and reliable quality assurance in biobanks, a quality management system (QMS) is indispensable. Through implementation of standardized procedures and processes, the QMS guarantees correct treatment of the biospecimens and thus ensures high quality of both biospecimens and associated data [6,7]. In Germany, many biobanks additionally strive for an accreditation according to DIN EN ISO 20387:2018 besides already existing DIN EN ISO 17020:2012 and DIN EN ISO 17025:2018 accreditation or ISO 9001:2015 certification to adapt their quality systems to international standards.

During the accreditation process, biobanks need to prove the competence of their employees through training certificates. However, existing national as well as international education and training programs predominantly address biobank managers, project managers, or biobank personnel in general [8]. The technical staff in biobanks, responsible for quality-assured collection, storage, processing, and release of biospecimens, however, is neglected by existing national education measures. This is not only true for technicians in biobanks but very similar for many technicians in research institutions in general in Germany. The diverse range of continuing education courses offered by universities usually contains few to no courses for topics relevant to technical staff in laboratories in general. For this reason, technicians at the Max Delbrück Centre in Berlin and at the Goethe University in Frankfurt, for example, have come together to form an interest group in order to make their voices heard and to exchange knowledge within their network [9]. On a different level, the German Institute for Continuing Education for Technologists and Analysts in Medicine is a registered association (https://diw-mta.de, accessed on 14 April 2021) amongst others for technical personnel in general, offering extra-occupational training courses with the focus on senior executive staff. These courses cost several thousand Euros, last between 18 and 36 months, and cover areas such as laboratory medicine, radiology technology, and systems management, and thus are out of scope for many technicians.

Considering the importance of the technical staff in biobanks for correct handling of biospecimens, and therefore biospecimen quality, the German Biobank Alliance (GBA), a national network of 20 biobank sites and one IT development center, aims to develop a holistic education program on biobank-specific topics for this neglected group of employees. A holistic education program based on common learning objectives could pave the way for additional harmonization of processes by providing a missing link. Subsequently, this education program could further harmonize biobank processes and deepen the understanding of why certain standard operating procedures have to be followed. This will lead to more comparable biospecimens quality across the different national biobank locations, ultimately contributing to sustainable reproducible research results.

In order to tailor the educational content and the provision specifically to the needs of technical personnel in biobanks, their needs and opinions had to be collected to answer the following questions:(A)What are the educational needs of the technical personnel in biobanks?(B)What learning objectives can be inferred from these needs for the education program for technical staff in biobanks?

## 2. Materials and Methods

**Questionnaire Development:** Due to the lack of an established questionnaire fitting the authors’ needs, the GBA education and training working group developed a 55/56-items online survey to assess the educational needs of technical personnel in German biobanks (see Appendix A). The questionnaire was divided into five main parts covering the areas of relevant demographics, educational needs, existing knowledge, previous participation in advanced trainings, and questions along the design of a future advanced training for biobank technicians.

Part one contained four questions covering the relevant position in the biobank, education and working experiences. A filter question split the remaining questionnaire into two different sections: one for the coworkers of the technical personnel (Appendix A) and one for the technical personnel (Appendix A) itself. The respective questions were comparable in terms of content but linguistically tailored to the relevant test group. The second part of the questionnaire was designed to identify the educational needs of the technical personnel in biobanks. Using 15 questions, the respondents were asked to grade their own educational needs or to estimate the needs of the technical personnel concerning different biobanking areas, respectively. As these above-mentioned biobank topics were very broad and thus unspecific, they were not sufficient for the formulation of detailed learning objectives. Therefore, each question contained an additional free text section where survey participants could state which topic of the different presented biobank areas in particular were of most importance for them.

The 20 questions that followed in the third part aimed for identification of existing knowledge of the technical personnel regarding specific aspects of biospecimen handling. In part four of the questionnaire, ten questions were asked to assess quantity, topic, location, costs, and general experience with previously attended training courses. The fifth and last part of the questionnaire contained eight (technical personnel) or nine questions (coworkers of the technical personnel) regarding existing budget for advanced trainings, preferred studying techniques, and preexisting experience with online educational tools. In the end, the participants were asked to specify their age group and were given the opportunity to add general concerns regarding the survey as free text.

The questionnaire was pre-tested with twelve biobank employees (seven technicians and five coworkers of the technical personnel) from three different biobank locations and modified accordingly. Layout and design of the online questionnaire was determined by the online survey software SurveyMonkey (SurveyMonkey, Portland, OR, USA).

**Quantitative data collection and statistical analysis:** The questionnaire was sent to biobank directors/managers of the German Biobank Alliance (in 2017 by the time the survey was conducted GBA comprised eleven biobank sites) and to other national biobanks using the e-mail list of the working group on biobanking of the research umbrella organization “Technology, Methods, and Infrastructure for Networked Medical Research” (TMF) (*n* = ~200) with the request to forward the survey to the identified target personnel, which included both the technical personnel and their direct coworkers (chain sampling). We decided to use chain sampling, as the technical personnel in German biobanks are not organized in e-mail lists and there is a high staff turnover. Therefore, chain sampling represented the best opportunity to contact as many technical personnel as possible.

The data were collected within a three-week period in November 2017. Reminders were sent twice—after two weeks and two days before the survey was closed. No incentives were offered. Subsequently the relevant data were downloaded from SurveyMonkey and analyzed with Excel. Descriptive statistical analysis was conducted. As the vast majority of answers were voluntary, results of participants who did not answer all questions but completed the questionnaire were included.

**Qualitative data collection:** To substantiate the findings of the survey, the results were analyzed and subsequently discussed in a face-to-face workshop with 16 technicians from nine different GBA biobank locations. In this workshop, the technicians were asked to state in which distinct subjects they would have liked to have had advanced training. Topics for commenting included (1) entry, (2) processing, (3) storage, and (4) release of biospecimens. For this purpose, index cards were distributed, and the technicians were asked to write down as many subjects, methods, etc. concerning the abovementioned four biobank topics as they wanted. Afterward, the written index carts were collected, displayed, and clustered together with the participating technicians.

**Data analysis and identification of learning objectives:** Descriptive statistical analysis was conducted. The majority of the presented questions of the questionnaire used a 7-point rating scale with only the lowest and highest scores specifically defined. Each scale was assigned a number for calculation of the arithmetic mean and standard deviation (SD). The results of the face-to-face workshop and the online survey (including both the results of the technicians and their coworkers) were used to formulate learning objectives for biobank technicians.

This was done in a process that resembled analysis methods such as qualitative content analysis [10] and thematic analysis [11]. The latter had to be adapted to the specific aim of formulating learning objectives. The process to formulate learning objectives used by the authors is illustrated in Figure 1 and consisted of six steps: collection of topics, reduction to key message, limitation to six most frequent biospecimen types in GBA biobanks (if applicable), addition of objective taxonomy in the cognitive area (level of expertise) according to Bloom [12], discussion of learning objectives, and finalization. A first set of learning objectives based on the answers from the survey and the workshop was discussed and validated by the authors—all experienced biobank staff with different backgrounds—in several telephone conferences and two face-to-face workshops. Subsequently, the learning objectives were thoroughly structured and sorted according to the biobank workflow.

**Ethics and data protection:** Each survey participant was asked to give his/her informed consent by clicking a mandatory box on the first survey page. Here, relevant information on the study was provided, including the purpose of the study and data handling/data protection. No cookies or IP-addresses were used to track the responses of individual participants. Correspondingly, participants were able to start the questionnaire several times from the very beginning. This effect was intentional, as the majority of technical biobank employees did not possess an individual computer at their workspace but rather shared a device with several coworkers. However, to avoid the risk of including multiple answers of one participant who started the survey several times, only completed questionnaires were considered for data analysis. Participants of the workshop provided consent orally after being informed about the purpose of the workshop.

## 3. Results

### 3.1. Survey Demographics

In total, 79 people completed the online survey. Of the 79 participants 54% defined themselves as being technicians (*n* = 43), 15% as biobank directors (*n* = 12), 14% as project leader/biobank managers (*n* = 11), 9% as quality managers (*n* = 7), 5% as laboratory managers (*n* = 4), and 3% as IT coworkers (*n* = 2) (Figure 2A). The answers of the coworkers of the technicians (biobank directors, project leader/biobank manager, quality manager, laboratory manager, IT coworker) were summarized (*n* = 36) and compared with the answers of the technical personnel (*n* = 43). The response rate could not be calculated as chain sampling was used. In the survey, the technicians were asked about their professional experience in biobanking (Figure 2B). Approximately 21.4% of the technicians stated that they had been working for less than a year (*n* = 9), 38.1% for one to under three years (*n* = 16), 19.1% claimed they had been working for three to under five years in a biobank (*n* = 8). The remaining 21.4% declared that had been working in a biobank more than five years. According to their reply, 60% of the technicians had been working in a biobank for less than three years (Figure 2B).

### 3.2. Educational Needs of the Technical Personnel

To identify the educational needs of the biobank technicians, the technicians themselves (Figure 3A) were asked to state their interest or lack of interest in training courses in seven biobank areas presented using a 7-point rating scale. Additionally, the coworkers were asked to indicate how many of their technicians they thought would need training in the same areas, using a 5-point rating scale (Figure 3B). Based on the answers to these questions and the corresponding calculated average, the biobank areas were sorted in descending order, with the most frequent topic being considered most relevant from the participants’ perspective. The technical staff evaluated the biobank areas “storage of biospecimens” (mean = 5.14; DS = 1.91), “quality management” (mean = 5.12; SD = 1.82), and “ethical, legal, and social issues” (ELSI) (mean = 5.05; SD = 1.90) as most important for advanced trainings, followed by “entry, processing, and release of biospecimens” (mean = 4.9; SD = 2.18) and “data management/data protection” (mean = 4.8; SD = 2.02).

The coworkers of the technicians ranked ELSI in first position (mean = 3.45; SD = 1.48), followed by the topics “quality management” (mean = 3.42; SD = 1.37) and “data management/data protection” (mean = 3.24; SD = 1.35).

Both groups, the technical personnel as well as their coworkers ranked the topics “occupational safety” (technical staff: mean = 4.46; SD = 1.99/coworker: mean = 2.35; SD = 1.39) and “collection and transport of biospecimens” (technical staff: mean = 4.26; SD = 1.93/coworker: mean = 2.74; SD = 1.36) as the least interesting subjects for a potential advanced training.

Apart from prioritizing the presented biobank areas, the survey participants were also asked to write down specific topics related to the categorized areas that would be particularly important for them to learn in advanced training (Appendix A).

To allow for transparency regarding the process of developing learning objectives, an example on how the participant’s contributions were integrated into the resulting learning objective is provided here: When asked which topics were most important to them in terms of “retrieval and transportation of biospecimens”, the technical staff stated “Finding optimal transport conditions and documentation for the various material types (confounding factors, e.g., CO_2_ influence, N_2_ without crystal formation)”, “Transporting nitrogen containers”, and “At what temperatures do the different samples have to be shipped?” (workshop and survey answers). Their coworkers declared “Parameters that should be considered from the time of removal; sources of error” as well as “Avoid thawing of the sample”. These quotations were abstracted and thus reduced to their basic message, which in these examples refer to the optimal transport conditions of biospecimens. As optimal transport conditions cannot be discussed in one training course for all potential biospecimens, it was decided to focus on the six most common types of biospecimens stored in GBA biobanks, which are tissue samples, urine, serum/plasma, whole-blood, RNA/DNA, and peripheral blood mononuclear cells (PBMCs). As a result, the learning objective 2.1 was defined as follows: “The technicians can elucidate the optimal conditions for transportation of the six most common types of biospecimens”. Based on the free text answers of both the technical personnel as well as their coworkers in the survey (Appendix A) and the results of a face-to-face workshop, 41 learning objectives representing the basis for an advanced training program for the technical personnel in biobanks were inferred (Table 1). The answers of the coworkers were also considered for the learning objectives, as they were capable of judging the importance of certain biobank topics for advanced education as well. The resulting learning objectives were grouped into nine categories and sorted along a biospecimen workflow starting from the collection of the biospecimen from the patient/sample donor to the transfer to the inquiring researcher (Figure 4). A detailed list of all 41 learning objectives can be found in Table 1.

The last part of the survey was prepared to obtain information on how to design the basic framework of the advanced training program. Therefore, the technical personnel as well as their coworkers were asked to state their preferred learning method. According to survey results, a vast majority (72%, *n* = 31) would prefer a combination of theoretical and practical aspects, followed by a preference for practical training only (14%, *n* = 6) (Figure 5).

To decide whether further education and training for biobank technicians could be offered in English (adoption of existing courses) or if training has to be in German, the technical personnel were asked to assess their English skills. The distribution in the pre-defined categories was (a) “not speaking English at all “(0%), (b) “my English is rather weak” (25.6%, *n* = 11), (c) “I can communicate in English” (48.8%, *n* = 21), (d) “I can communicate easily in English” (25.6%, *n* = 11), and (e) “I am a native speaker” (0%). The results indicate that approximately three-quarters of the technicians might not be capable of comprehending complex biobank-specific content instantly.

## 4. Discussion

**Contextualizing the findings:** So far, there is no specific national education program for the technical staff in biobanks. In this survey, we identified that the work experience of technical personnel is on average less than three years. This limited work experience and the missing educational options for technical personnel on a national level implies that there is a clear need for advanced training of this type for biobank employees. According to the survey results, up to 40% of the technical personnel had previously participated in advanced trainings. However, the overwhelming majority of these trainings, in total 82.9%, were on biobank independent topics such as general work safety (28.7%) or on scientific events such as symposia (25.7%). Almost none of these events were on procedures that related to direct handling of samples.

Around the globe, there are increasing efforts to develop training courses for (future) biobank employees. For example, the Medical University of Graz offers several postgraduate courses as part of a master’s program in biobanking, which can be taken in parallel with a full-time job. In addition, practical on-site trainings on relevant biobank specific topics are organized [13]. Similarly, the Catholic University of Lyon offers a master’s program in Biobanking [14]. The University of Luxembourg offers a 3-week course involving the Integrated Biobank of Luxembourg [15]. The Canadian Tumor Repository Network [16], as well as the University of British Columbia [17], offer biobanking online courses. The University of Cairo offers a short course [18]. Furthermore, the Central South University in China offers a one semester course on cryobiology and biobankology [19]. A good overview of these activities can be found here [8]. Unfortunately, for the majority of the surveyed technical personnel enrolling in an English course abroad would not be an option. Most of the technicians are not sufficiently qualified for the courses because they are predominantly aimed at postgraduates and the course language is English, which is a high barrier for most technical employees. According to the survey results, only a one-quarter of the interviewed technicians would be able to follow or discuss technical content in English. This result also indicates that the majority of technicians cannot inform themselves on the detailed specifics regarding the optimal handling of samples, as those are predominately published in English (e.g., ISBER Best Practices).

In summary, the obtained survey results underline the necessity of training programs for the technical staff of biobanks to be conducted in German. As the overwhelming majority of the surveyed technical personnel was highly motivated to take part in various training courses, it is very likely that a training program based on the outcome of the presented survey would match the exact interests of this target group.

In order to define the content of this training program the technical staff, as well as their coworkers, were surveyed concerning specific biobanking subjects. The resulting answers show a slightly different prioritization of the topics by the two participating groups (Figure 3A,B), which could be an effect of the different wording of the question. The technical personnel were asked to state the biobank topics for which they would like to have training courses, whereas their coworkers were asked to indicate how many of their technical personnel would benefit from advanced training in certain topics. The answers of the technical personnel may potentially have differed if they had been asked to indicate which topics would be most essential. However, quality management was ranked second highest for both groups, which is in line with what education experts [20] see as most relevant topic.

Furthermore, the differences in the answers of the technical personnel and their coworkers might also be caused by the different perspectives of both groups. It is most likely that the director of a biobank might estimate the importance of certain topics for the biobank differently. However, the overall prioritization of the different biobank topics by the technical personnel and their coworkers was comparable. The only biobank topic that was strikingly differently weighted by the technical staff compared with the coworkers was the topic “storage of biospecimens”. This topic was highly favored by the technical staff, resulting in a first-place ranking in their survey, whereas their coworkers only ranked it in the third to the last position. One possible explanation for this discrepancy could be a poor communication between the technical personnel and those at the management level of a biobank. It is conceivable that the biobank managers and directors, who represented 64% of the participating coworkers, could not really assess the need for advanced trainings of their technical personnel because the biobank directors/managers were not aware of the day-to-day routine of personnel staff.

**Limitations**: The response rate of the survey could not be calculated as we used chain sampling to reach out to the technical personnel and their coworkers. However, just 79 out of 142 participants or 55.6% finished the survey. As neither tracking of IP addresses nor cookies were used, we could not identify individual responses and were thus not able to ascertain if the termination rate of 44.4% was an effect of participants who started the survey several times. Due to the same reason, we also could not exclude the possibility that some persons completed the survey twice. Nevertheless, we do not consider this scenario very likely, as the participants themselves were interested in sound results.

The answers of the technical personnel and their coworkers concerning the survey questions used to identify the educational needs of the technical personnel (Figure 2A,B) were not statistically comparable, as both groups received different scales (seven items in case of the technical personnel and five items in case of the coworkers) and the coworkers were asked to judge the educational needs of all their technicians. In addition, wording of the respective questions was slightly different between both surveyed groups.

**Future implementation of the learning objectives:** The learning objectives cover the necessary knowledge that every biobank technician should have in order to correctly fulfil his or her daily work in accordance with the various regulations.

The developed learning objectives, as well as the other survey results, serve as a basis for GBA for the development and implementation of an advanced training program for technical staff in biobanks. The training program itself will be divided into a theoretical part delivered in online courses and a practical part delivered in subsequent practical trainings on the respective topic. According to the survey results, this combination also represents the most preferred advanced training format. The online modules are intended to cover a time frame of two to four hours each, depending on the scope and complexity of the respective topic and technique. As biobank directors are often not willing to release their technical staff for training purposes for a longer time, the practical units of the training will be kept rather short, i.e., in a range of two days, including travel. These practical training sessions could take place several times a year on different techniques. During the implementation of the GBA education program, at a later stage, it is also planned to include the more experienced technical personnel, e.g., in terms of extra trainings on specific techniques or new findings in the field of biobanking. Until then, the different levels of experience of the participating technical staff will be addressed, for example, in such a way that more advanced technical staff in a certain procedure or technique will be actively involved in teaching and mentoring the participating beginners. This allows the advanced technicians to confirm and reflect their own knowledge and skills, as well as to receive appreciation for their contribution to educate others.

We see two options to reduce the workload for GBA when developing and implementing the envisaged training program. On the one hand, a collaboration with partner organizations would be helpful. Being a member of Biobanking and Biomolecular Resources Research Infrastructure—European Research Infrastructure Consortium (BBMRI-ERIC), it would of course be conceivable to identify partner biobank networks with which to develop a joint training program for technical personnel in German-speaking countries, i.e., together with the Swiss and Austrian biobank networks, for example. In general, there are many different working groups on BBMRI’s level that are potential partners for development of an international training program for the technical personnel. On the other hand, it is possible to search for potential cooperation partners that already designed an advanced training program for biobank employees [17] that could perhaps be adapted to the needs of the German technical personnel. For example, the Office of Biobank Education and Research (OBER) of the University of British Columbia has developed a comprehensive course on biobanking, which could be customized for the German context. A prerequisite for this approach would be that the identified learning objectives match the content of the already existing course.

## 5. Conclusions

Advanced education of the technical personnel in biobanks is essential for harmonizing procedures related to direct handling of samples in order to assure their comparably high quality across different biobank locations. As technical biobank personnel are currently neglected by (inter)national advanced training options, the GBA has defined the learning objectives of a potential advanced training program for this group of biobank staff.

These learning objectives and survey results could be used by other national nodes within BBMRI-ERIC or other biobank networks beyond Europe to develop advanced training for biobank technical staff, thus, contributing to a further harmonization at the root of biospecimens and data handling.

## Figures and Tables

**Figure 1 biotech-10-00007-f001:**
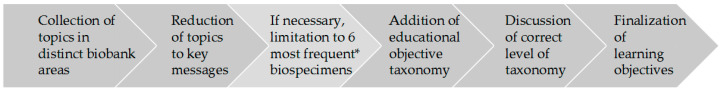
Flow chart of the process of the development of learning objectives. The * indicates the six most frequent biospecimens stored in partner biobanks of the German Biobank Alliance.

**Figure 2 biotech-10-00007-f002:**
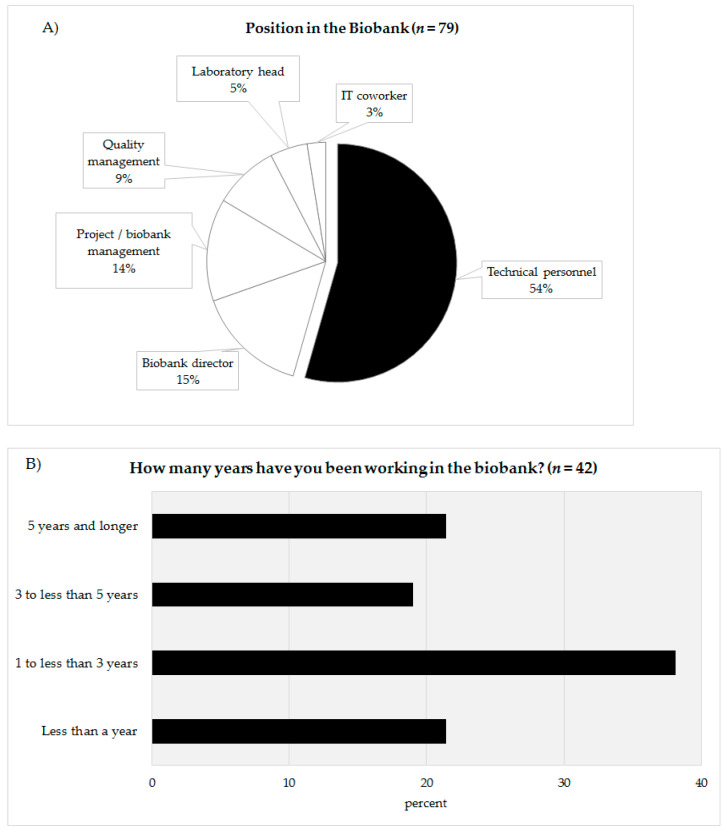
(**A**) Categorization of the survey participants according to their position in the biobank. (**B**) Work experience in years of technical personnel in biobanks.

**Figure 3 biotech-10-00007-f003:**
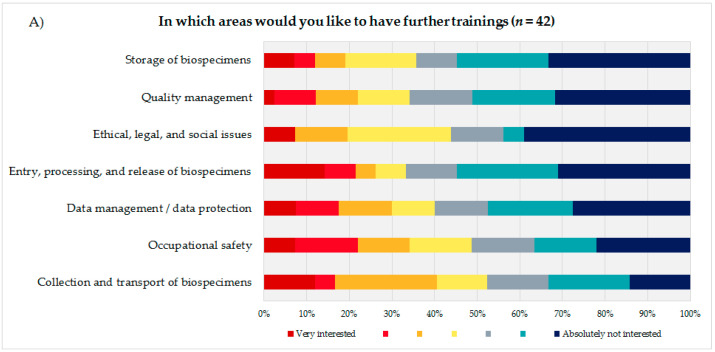
(**A**) Evaluation of seven presented biobank topics by the technical personnel. (**B**) Estimation of the coworkers of the technical personnel on how many of their technicians would need advanced training in the presented biobank topics.

**Figure 4 biotech-10-00007-f004:**
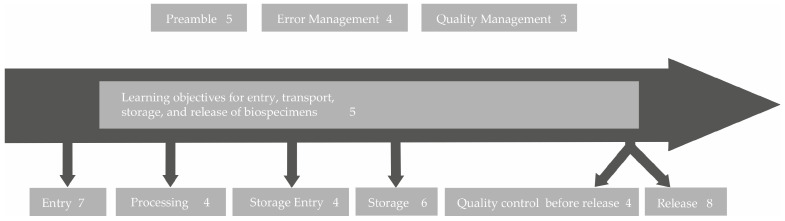
A sorting of the identified learning objectives along the typical biospecimen life cycle within a biobank. The numbers behind the topics are the numbers of learning objectives generated in this area.

**Figure 5 biotech-10-00007-f005:**
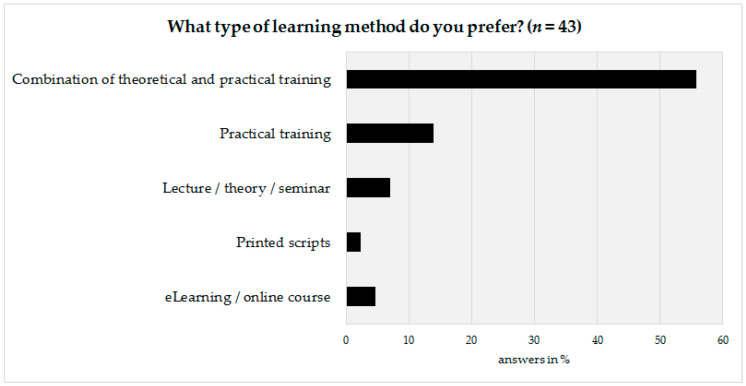
Preferred learning method of the technical personnel.

**Table 1 biotech-10-00007-t001:** List of the 41 learning objectives, sorted according to nine main biobanking categories.

No	Learning Objectives
0	**Preamble**
0.1	The technicians can constitute three advantages of storing biospecimen in the local centralized biobank to other clinical personnel.
0.2	The technicians can explain the difference between pseudonymization (de-identification with link) and anonymization (de-identification without link).
0.3	The technicians can illustrate basic aspects of ethical and legal issues for acquiring, storing, and releasing biospecimen.
0.3.1	The technicians can report the general content of an informed consent and can describe how to act according to the donor’s will.
0.3.2	The technicians can explain what needs to be done with biospecimen in case of a withdrawal of the informed consent.
1	**General aspects of entry, transport, storage, and processing of biospecimen**
1.1	The technicians can list pre-analytical factors that influence the quality of solid and liquid biospecimen.
1.1.1	The technicians can name factors that negatively influence the stability, integrity, and concentration of RNA and DNA.
1.1.2	The technicians can name quality assurance aspects for the six most common types of biospecimen (tissue, urine, serum/plasma, whole blood, RNA/DNA, PBMC).
1.1.3	The technicians can evaluate the biospecimen quality based on GBA quality parameters.
2	**Entry of biospecimens**
2.1	The technicians can illustrate how to label a sample in an appropriate way (different ways of labelling, e.g., adhesive labels, labelling by handwriting, barcodes).
2.2	The technicians can explain and apply the most important aspects of data protection regarding the labelling of biospecimen.
2.3	The technicians can explain mandatory aspects of documentation concerning the biospecimen entry.
2.3.1	The technicians can explain “cold and warm ischemia time” and why these factors are critical for tissue quality.
2.3.2	The technicians can elucidate which processing times are relevant for the processing of blood and why these times are critical for the biospecimen quality.
2.3.3	The technicians can explain the HIL-index.
3	**Processing of biospecimens**
3.1	The technicians can name the optimal processing temperatures (until the biospecimen storage entry) for the six most common types of biospecimen (tissue, urine, serum/plasma, whole blood, RNA/DNA, PBMC).
4	**Storage Entry**
4.1	The technicians can depict the single steps of storage entry with regard to the six most common types of biospecimen (tissue, urine, serum/plasma, whole blood, RNA/DNA, PBMC).
4.2	The technicians can explain which method of freezing is most appropriate for which of the six most common types of biospecimen (tissue, urine, serum/plasma, whole blood, RNA/DNA, PBMC).
4.3	The technicians can describe the safety measures that have to be applied when transporting biospecimen using liquid nitrogen.
4.4	The technicians can apply the safety measures when transporting biospecimen using liquid nitrogen (implementation within the on-Site Training #2).
5	**Storage**
5.1	The technicians can depict advantages and disadvantages of different storage equipment for the six most common types of biospecimen (tissue, urine, serum/plasma, whole blood, RNA/DNA, PBMC).
5.2	The technicians can depict which factors influence the quality of biospecimens during long-term storage.
5.2.1	The technicians can explain the influence of the storage temperature on biospecimen.
5.2.1	The technicians can describe how they ensure that the biospecimen is not exposed to temperature fluctuations during transport from one storage area to another.
6	**Release**
6.1	The technicians can explain the framework prerequisites that have to be fulfilled for releasing biospecimen.
6.2	The technicians can describe the process of releasing biospecimen.
6.2.1	The technicians can describe how the six most common types of biospecimen need to be handled for transport to ensure the biospecimen quality (tissue, urine, serum/plasma, whole blood, RNA/DNA, PBMC).
	(a)–The technicians can describe what should be added to the correctly packed parcel for quality reasons.
	(b)–The technicians can explain the correct identification mark of the packaging for the six most common types of biospecimen for transport/shipping (tissue, urine, serum/plasma, whole blood, RNA/DNA, PBMC).
7	**Analysis before release**
7.1	The technicians can depict the thawing protocols of the six most common types of biospecimen (tissue, urine, serum/plasma, whole blood, RNA/DNA, PBMC) with regard to the planned biospecimen usage.
7.2	The technicians can list what kind of biospecimen is suitable for the most common types of analyses.
7.2.1	The technicians can explain histological and immunohistochemical techniques for tissue processing.
7.2.2	The technicians can outline (and perform) the process of generating tissue slices.
8	**Error Management**
8.1	The technicians can describe what to do when the wrong label or no label is attached to a biospecimen sample tube.
8.2	The technicians can give at least one recommendation to avoid errors when entering the six most common types of biospecimen (tissue, urine, serum/plasma, whole blood, RNA/DNA, PBMC) samples into storage.
8.3	The technicians can explain how to avoid disturbing factors (e.g., ice formation when storing in liquid nitrogen and the entry of carbon dioxide into the biospecimen sample) when entering and retrieving biospecimen into/from the store.
8.4	The technicians can give at least one recommendation to avoid errors when retrieving the six most common types of biospecimen (tissue, urine, serum/plasma, whole blood, RNA/DNA, PBMC).
9	**Quality Management**
9.1	The technicians can explain general aspects and approaches of a quality management system for biobanking.
9.2	The technicians can describe how Standard Operating Procedures (SOPs) for storage entry, processing, storage, release, and transport of biospecimen are structured.
9.3	The technicians can describe and execute the emergency plan for biospecimen storage.

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
