# Peer review of "Harmonization of Biobank Education for Biobank Technicians: Identification of Learning Objectives"

_biotech, 2021, doi:10.3390/biotech10020007_

Round 1
Reviewer 1 Report
Thank you for your manuscript, it is well-structured and the results are described clearly. The following minor revisions are recommended in order to strengthen the contextual position of your arguments:
- The need for further educational opportunities for technical staff is clearly presented, however it is not clear what this opportunity might look like in the German context. Could the authors provide 1-2 examples from other technical training opportunities (if possible in laboratories, in Germany) so that the readership understands that this has been achieved before, albeit in other fields.
- As 60% of technicians working in biobanks were for less than three years, is there a possibility that there need to be two training sets, one for entry level and one for more experienced staff (re-training)? It would be good if the authors could comment on this, 1-2 short sentences would suffice.
- How do your findings compare against other laboratory-based training course needs in Germany? The contextual description is very clear for biobanking, but it is not well-described if these findings align with previous findings in related fields or if this is simply not known - and what was done in those cases. A few short sentences with appropriate references would be sufficient.
- The reason for asking the point immediately above, is because one of the possibilities of designing a course, is to do so in collaboration with an organisation that already exercises an educational program in a related field [e.g. i) the ISBER Biorepository certification for biobanking staff has been created in collaboration with the American Society for Clinical Pathology (ASCP); ii) the course at the Catholic University of Lyon was developed with substantial input from IARC/WHO]. While it is understandable that the customization to the German context might preclude a wide-reaching international collaboration (e.g. with ASCP or IARC/WHO), it is not clear if such a collaboration would be feasible within the German context (e.g. GBA and another partner). A few short sentences would allow the readership to understand the arguments expressed within a wider context.
- As the authors point out, existing educational opportunities at biobanking range from webinar series, to a 3 week course, to a university degree. According to the authors' work, what would be the optimal timeframe within the German context and why? Sometimes there are local restrictions and regulations for such courses (e.g. CPD hours) and sometimes not, as such a succinct comment would be sufficient and beneficial.
Author Response
Dear reviewer,
thank you for your valuable feedback to our manuscript. We have intensely worked on your suggestions for improvement and addressed them as described in the table below. We used the track change mode in Word to indicate changes directly in the manuscript as well.
| Reviewer | Comment | Answer |
| 1 | The need for further educational opportunities for technical staff is clearly presented, however it is not clear what this opportunity might look like in the German context. Could the authors provide 1-2 examples from other technical training opportunities (if possible in laboratories, in Germany) so that the readership understands that this has been achieved before, albeit in other fields. | We added a few sentences to describe the situation of further educational opportunities für technical staff in Germany in the introduction. |
| As 60% of technicians working in biobanks were for less than three years, is there a possibility that there need to be two training sets, one for entry level and one for more experienced staff (re-training)? It would be good if the authors could comment on this, 1-2 short sentences would suffice. | Designing the learning objectives, we did not initially distinguish between different levels of experience of the technical personnel. These learning objectives were meant to cover the basic knowledge concerning biospecimens handling that each technician in a biobank should have. However, the technical personnel indeed possess very different educational backgrounds and we would like to expand the learning objectives in order to satisfy technicans that are very experienced, for examples in trainings of very specific and complex topics. A comment to this remark is implemented in the disussion section. | |
| How do your findings compare against other laboratory-based training course needs in Germany? The contextual description is very clear for biobanking, but it is not well-described if these findings align with previous findings in related fields or if this is simply not known - and what was done in those cases. A few short sentences with appropriate references would be sufficient. | We added a couple of sentences in the introduction to describe the situation of further educational possibilities for technicians in German in general and describe what they do about it. Regarding the references, we were only able to find an article on this topic in German and included it. | |
| The reason for asking the point immediately above, is because one of the possibilities of designing a course, is to do so in collaboration with an organisation that already exercises an educational program in a related field [e.g. i) the ISBER Biorepository certification for biobanking staff has been created in collaboration with the American Society for Clinical Pathology (ASCP); ii) the course at the Catholic University of Lyon was developed with substantial input from IARC/WHO]. While it is understandable that the customization to the German context might preclude a wide-reaching international collaboration (e.g. with ASCP or IARC/WHO), it is not clear if such a collaboration would be feasible within the German context (e.g. GBA and another partner). A few short sentences would allow the readership to understand the arguments expressed within a wider context. | As the focus of this work was to identify the learning objectives for a training program for the technical staff in biobanks in Germany, we did not go into detail about the implementation in a training program. However, since we also believe that it is of interest to the readership how the training program could be designed and which partner would be conceivable, we have included a corresponding text passage in the discussion section. | |
| As the authors point out, existing educational opportunities at biobanking range from webinar series, to a 3 week course, to a university degree. According to the authors' work, what would be the optimal timeframe within the German context and why? Sometimes there are local restrictions and regulations for such courses (e.g. CPD (continuouing professional development) hours) and sometimes not, as such a succinct comment would be sufficient and beneficial. | The optimal time frame of a training was discussed in several workshops with the technical staff. When asked, they mostly indicated that they would prefer an on-site training of about a week, not only to have time to understand the topic better, but also to have time to network. However, there is some reluctance among biobank directors to release technical staff for training for a longer period at a time. This is due to the fact that a lot of practical work in a biobank depends on the technical staff, which cannot be easily postponed. Therefore, it is likely that technical staff cannot be released for a week or more to attend practical on-site training. In addition, there is no legal obligation for technical staff to attend further training in order to exercise their profession. For the same reason, there are also no specifications on how further training for technical staff should be designed. In order to create a training program for the technical staff that is also convenient for the biobanks, we have decided to offer short practical trainings, around 2-days including the travel to and from the biobanks, that focus on only one topic or method. The practical training will be accompanied by suitable online education modules that will deliver the theoretical context of the training. An exchange of questions by the technical personnel will be achieved by an online discussion forum which will be supervised by biobank experts. A comment to this remark was also implemented in the discussion section. |
Reviewer 2 Report
Interesting diagnostic about the preparedness of the biobanks personnel in Germany. Despite a low number of respondents, the panorama obtained is enough to concern and propose the measures to alleviate the situation. Possibly the results would be enhanced with a more active attitude of biobanks personnel and more strict control of responses to avoid imprecisions regarding double responses or incomplete responses.
Author Response
Dear reviewer,
thank you for your valuable feedback to our manuscript. We have intensely worked on your suggestions for improvement and addressed them as described in the table below. We used the track change mode in Word to indicate changes directly in the manuscript as well.
| Reviewer | Comment | Answer |
| 2 | Interesting diagnostic about the preparedness of the biobanks personnel in Germany. Despite a low number of respondents, the panorama obtained is enough to concern and propose the measures to alleviate the situation. Possibly the results would be enhanced with a more active attitude of biobanks personnel and more strict control of responses to avoid imprecisions regarding double responses or incomplete responses. | This study included one of the largest surveys among technical staff in biobanks. The survey was advertised in all major national biobank networks as the “Kompetenznetze” or “AG Biobanken of the TMF e.V.”. However, the design of the survey indeed does not allow exclusion of duplicated answers, as we did not track cookies or IP-addresses, which potentially distorts the results of the survey. This effect was intentional, as the majority of technical biobank employees does not possess an individual computer at their workspace, but rather share a device with several co-workers. Thus, a limitation of one survey answer per IP-address would have had the effect, that only one technician per biobank would have had the chance to respond to the survey. In order to better assess the participation rate of technical staff, we have tried to assume the total number of technical staff in academic biobanks. In Germany 39 academic medical centers exist. If we assume that each of these universities possess a biobank with 5 technical personnel, a number that was identified in our survey, we end up with approximately 195 technical personnel. If the number of 43 technical staff who participated in the survey is related to the number calculated above, it can be assumed that about a quarter of the national technical staff in academic biobanks participated in the survey. This figure is not extremely high, but nevertheless gives a valid assessment of the situation of technical staff in biobanks |
Reviewer 3 Report
The article by Hartung and coworkers describes the results of a web-based survey which was specifically developed and then sent to all german biobanks to assess the educational needs of the biobank technical personnel. The 56-items survey was completed by 79 biobank employees, including 43 technicians. The results were analyzed and discussed within a face-to-face workshop with 16 technicians from e different national biobanks. The analysis led to the identification of 41 learning objectives that could be exploited in specific training courses for biobank technicians in order to improve harmonization and sustainability of biospecimen quality
The article is well presented and addresses an important issue in biobanking. The Introduction reports the relevant references and open issues in the field. The Methods and the performed analysis are generally well described and the findings are interesting.
I would suggest: (i) adding the percentage of personnel that completed the survey with respect to the total, to have an idea of the representativeness of the results collected, and (ii) taking more care and implementing the description of how the learning objectives were determined, that is the core of the article. Specifically, I would suggest describing better the procedure followed not just in the Methods, but also in the Results, moving also here the description of the specific example reported in the Methods (line 150). Moreover, I would recommend adding a general flow chart describing the entire procedure, which can be very useful for readers.
Can authors attach the questionnaire as supplementary material?
Finally, it would be valuable to discuss, based on the results and the learning objectives identified, some proposal of training courses to develop and/or offer to biobank technicians.
Minor points:
Figure 3 is not really self-explicative. Please add a Figure legend describing the scheme.
Table 1: I would suggest subdividing the Table into rows and columns and putting the principal categories (preamble…entry…..) in bold.
Author Response
Dear reviewer,
thank you for your valuable feedback to our manuscript. We have intensely worked on your suggestions for improvement and addressed them as described in the table below. We used the track change mode in Word to indicate changes directly in the manuscript as well.
| Reviewer | Comment | Answer |
| 3 | I would suggest: (i) adding the percentage of personnel that completed the survey with respect to the total, to have an idea of the representativeness of the results collected, and | As described in the limitations section of the paper in the discussion, in total 55,6 % (n=79) of the persons starting the survey (n=142) have completed it, resulting in a termination rate of 44,4%. Only answers of persons that have completed the survey were included in the different calculations in order to avoid potential duplication of answers, as it is very likely that some participants have started the survey a couple of times until finally finishing it. |
| (ii) taking more care and implementing the description of how the learning objectives were determined, that is the core of the article. Specifically, I would suggest describing better the procedure followed not just in the Methods, but also in the Results, moving also here the description of the specific example reported in the Methods (line 150). Moreover, I would recommend adding a general flow chart describing the entire procedure, which can be very useful for readers. | For a better understanding of the underlying method, we created an additional Figure for illustration of our workflow for the generation of learning objectives and expanded the text in the methods section including a new reference. In addition, as recommended, we moved the description of the specific example into the results section. | |
| Can authors attach the questionnaire as supplementary material? | Yes. The survey that aimed to assess the educational needs of the technical personnel is attached as supplemental table 1. It contains questions that were only addressed to the technical employees. Supplemental table 2 lists the questions for the co-workers of the technical personnel. | |
| Finally, it would be valuable to discuss, based on the results and the learning objectives identified, some proposal of training courses to develop and/or offer to biobank technicians. | We expanded the discussion to address this point. | |
| Figure 3 is not really self-explicative. Please add a Figure legend describing the scheme | The figure legend was expanded to be more self-explicable | |
| Table 1: I would suggest subdividing the Table into rows and columns and putting the principal categories (preamble…entry…..) in bold. | We adapted the layout of the table by setting the text orientation to left-justified and put the categories in bold, as in our original version of the table. |